# TESH-GCN: Text Enriched Sparse Hyperbolic Graph Convolutional Networks

## Abstract

Heterogeneous networks, which connect informative nodes containing semantic information with different edge types, are routinely used to store and process information in various real-world applications. Graph Neural Networks (GNNs) and their hyperbolic variants provide a promising approach to encode such networks in a low-dimensional latent space through neighborhood aggregation and hierarchical feature extraction, respectively. However, these approaches typically ignore metapath structures and the available semantic information. Furthermore, these approaches are sensitive to the noise present in the training data. To tackle these limitations, in this paper, we propose Text Enriched Sparse Hyperbolic Graph Convolution Network (TESH-GCN). In TESH-GCN, we use semantic node information to identify relevant nodes and extract their local neighborhood and graph-level metapath features. This is done by applying a reformulated hyperbolic graph convolution layer to the sparse adjacency tensor using the semantic node information as a connection signal. These extracted features in conjunction with semantic features from the language model (for robustness) are used for the final downstream tasks. Experiments on various heterogeneous graph datasets show that our model outperforms the state-of-the-art approaches by a large margin on the task of link prediction. We also report a reduction in both the training time and model parameters compared to the existing hyperbolic approaches through a reformulated hyperbolic graph convolution. Furthermore, we illustrate the robustness of our model by experimenting with different levels of simulated noise in both the graph structure and text, and also, present a mechanism to explain TESH-GCN's prediction by analyzing the extracted metapaths.

## 1 Introduction

Heterogeneous networks, which connect informative nodes containing semantic information with different edge types, are routinely used to store and process information in diverse domains such as e-commerce Choudhary et al. (2022), social networks Leskovec & Mcauley (2012), medicine Cohen (1992), and citation networks Sen et al. (2008). The importance of these domains and the prevalence of graph datasets linking textual information has resulted in the rise of Graph Neural Networks (GNNs) and their variants. These GNN-based methods aim to learn a node representation as a composition of the representations of nodes in their multi-hop neighborhood, either via random walks Perozzi et al. (2014); Grover & Leskovec (2016), neural aggregations Hamilton et al. (2017); Kipf & Welling (2017); Veličković et al. (2018), or Boolean operations Wu et al. (2020). However, basic GNN models only leverage the structural information from a node's local neighborhood, and thus do not exploit the full extent of the graph structure (i.e., the global context) or the node content. In the context of e-commerce search, based on a consumer's purchase of "[brand1] shoes", it is difficult to identify if they would also purchase "[brand2] shoes" or "[brand1] watch" merely on the basis of the products' nearest graph neighbors, however, global information on purchase behavior could provide additional information in identifying and modeling such purchase patterns. Analysis into such limitations has led to research into several alternatives that capture additional information such as hyperbolic variants Ganea et al. (2018); Chami et al. (2019) to capture the latent hierarchical relations and hybrid models Zhu et al. (2021); Yao et al. (2019) to leverage additional text information from the nodes

in the graph. In spite of their preliminary success, these aforementioned techniques fundamentally suffer from several critical limitations such as non-scalability and lack of robustness to noise in real-world graphs when applied in practice. Certain other attempts on aggregating a graph's structural information Ying et al. (2021) utilize graph metrics such as centrality encoding and sibling distance to show improved performance over other approaches. However, there is an exhaustive set of graph metrics and manually incorporating every one of them is impractical. Hence, practitioners need a better approach to automatically detect the most relevant graph features that aid the downstream tasks. For example, metapaths, heterogeneous paths between different nodes that preserve long-distance relations, are traditionally found to be good message passing paths in several graph problems Fu et al. (2020). However, they are only aggregated locally due to computational constraints, i.e., only a local k-hop neighborhood of a heterogeneous graph's node is considered while learning the metapaths. However, global metapaths can capture long-term relations between the nodes. To learn metapaths, we need to encode the path between two nodes and the semantic information contained in the path. Thus, The adjacency tensor of a heterogeneous graph[1] with a semantic signal can be used to extract both metapath information as well as aggregate local neighborhood features. *Efficiently encoding the entire adjacency tensor in training graph neural models can thus help capture all relevant metapath features.*

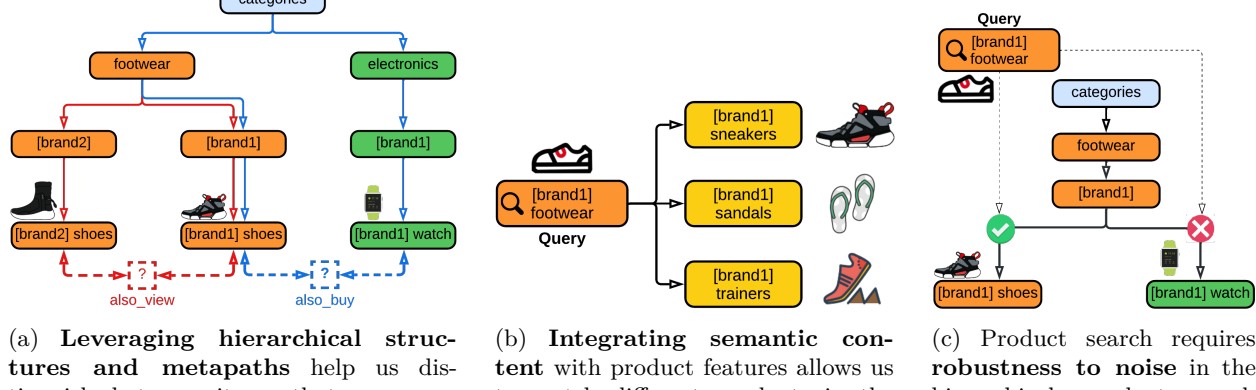

(a) **Leveraging hierarchical structures and metapaths** help us distinguish between items that are complementary (also_buy) or alternatives (also_view) of each other.

(b) **Integrating semantic content** with product features allows us to match different products in the catalogue with the query "[brand1] footwear".

(c) Product search requires **robustness to noise** in the hierarchical product graph structure caused by miscategorized items.

Figure 1: Challenges of graph representation learning in the E-commerce domain.

In addition to this, the nodes in the graph datasets also contain auxiliary information in different modalities (generally text) such as product descriptions in e-commerce graphs and article titles in citation networks. Such textual content can be encoded using popular transformer models Devlin et al. (2019), and consequently serve as an additional source of information. Thus, integrating these transformer models in the graph's representation learning process should improve the nodes' feature content during message aggregation and enhance the node representations. Recent hybrid graph-text based techniques Zhu et al. (2021); Yao et al. (2019) also attempt to integrate the node representations with semantic embeddings by initializing the node features with fixed pre-processed semantic embeddings. But, this does not completely leverage the representational power of transformer networks which can learn the task-specific semantic embeddings. Hence, we require a better approach that is able to focus both on the graph and text representation learning towards the downstream task. To summarize, in this paper, we aim to create a unified graph representation learning methodology that tackles the following challenges (examples from the e-commerce domain given in Figure 1):

1. *Leveraging metapath structures:* Existing GNN frameworks aggregate information only from a local neighborhood of the graph and do not possess the ability to aggregate graph-level metapath structures. However, graph-level information can aid in several graph analysis tasks where node's local neighborhood information is insufficient, e.g., in Figure 1a, we note that local node-level information is unable to distinguish between the relations of "also_buy" and "also_view", whereas, graph-level information allows us

---

[1]for a homogeneous graph, it will be a matrix

to do make the differentiation. Indeed, when attempting to combine information from the entire graph, existing methods suffer from over-smoothness Oono & Suzuki (2020). Moreover, the size of modern graph datasets renders aggregating information from the full graph infeasible.

2. *Incorporating hierarchical structures:* Most of the real-world graphs have inherent hierarchies, which are best represented in a hyperbolic space (rather than the traditional Euclidean space), for e.g., the product hierarchy shown in Figure 1a. However, existing hyperbolic GNNs Ganea et al. (2018); Chami et al. (2019) do not leverage the full graph when aggregating information due to both mathematical and computational challenges.

3. *Integrating textual (semantic) content:* Previous methods for integrating semantic information of the nodes are relatively ad-hoc in nature. For example, they initialize their node representations with text embeddings for message aggregation in the GNNs Zhu et al. (2021). Such methods fix the semantic features and do not allow the framework to learn task-specific embeddings directly from the nodes' original content, e.g., in Figure 1b, the product tokens "sneakers" and "sandals" are closer to the query token "footwear" in the e-commerce domain which is not the case in a broader semantic context.

4. *Robustness to noise:* Real-world graphs are susceptible to noise and hence require robust graph representation learning mechanisms, especially in the presence of multiple forms of data (i.e., graph structure and textual content), e.g., in Figure 1c, we observe that the task of product search is susceptible to noise in the product catalogue due to miscategorized items. Previous approaches do not leverage the complementary nature of graphs and text to improve robustness to noise in both of these modalities.

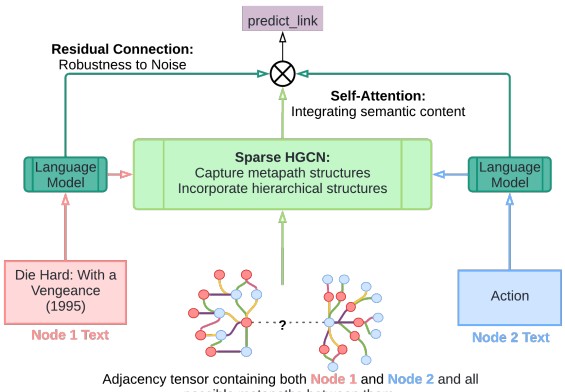

Figure 2: An overview of the proposed TESH-GCN model. The semantic signals are efficiently integrated with the nodes' local neighborhood and metapath structures extracted from the adjacency tensor.

To tackle the above challenges, we introduce *Text Enriched Sparse Hyperbolic Graph Convolution Network* (TESH-GCN), a novel architecture towards learning graph representations (illustrated in Figure 2) for the task of link prediction. In the case of heterogeneous graphs, the node adjacency information can be modeled as a tensor and can be used to both aggregate local neighborhood as well as extract graph-level metapath structures Fu et al. (2020). However, real-world adjacency tensors are extremely sparse ($\approx 99.9\%$ entries are zero)[2]. *TESH-GCN leverages the sparsity to efficiently encode the entire adjacency tensor and automatically captures all the relevant metapath structures.* We also utilize dense semantic signals from the input nodes which improve the model's robustness by making the representations conditional on both the graph and text information. To capture the semantic information of the nodes, we leverage the recent advances in language models Devlin et al. (2019); Feng et al. (2020) and jointly integrate the essential components with the above mentioned graph learning schemes. This allows nodes' feature content to be passed through the message aggregation and enhance performance on downstream tasks. In addition to this, our model's attention flow enables the extraction and comprehension of weighted inter-node metapaths that result in the final prediction. Summarizing, following are the major contributions of this paper:

---

[2]Sparsity ratios of our datasets are given in Table 2.

1. We introduce Text Enriched Sparse Hyperbolic Graph Convolution Network (TESH-GCN), which utilizes semantic signals from input nodes to extract the local neighborhood and metapath structures from the adjacency tensor of the entire graph to aid the prediction task.

2. To enable the coordination between semantic signals and sparse adjacency tensor, we reformulate the hyperbolic graph convolution to a linear operation that is able to leverage the sparsity of adjacency tensors to reduce the number of model parameters, training and inference times (in practice, for a graph with $10^5$ nodes and $10^{-4}$ sparsity this reduces the memory consumption from 80GB to 1MB). To the best of our knowledge, no other method has utilized the nodes' semantic signals to extract both local neighborhood and metapath features.

3. Our unique integration mechanism, not only captures both graph and text information in TESH-GCN, but also, provides robustness against noise in the individual modalities.

4. We conduct extensive experiments on a diverse set of graphs to compare the performance of our model against the state-of-the-art approaches on link prediction and also provide an explainability method to better understand the internal workings of our model using the aggregations in the sequential hyperbolic graph convolution layers.

The rest of this paper is organized as follows: Section 2 discusses the related work in the areas of link prediction and hyperbolic networks. Section 3 describes the problem statement and the proposed TESH-GCN model. In Section 4, we describe the experimental setup, including the datasets used for evaluation, baseline methods, and the performance metrics used to validate our model. Finally, Section 5 concludes the paper.

## 2 Related Work

In this section, we describe earlier works related to our proposed model, primarily in the context of graph representation learning and hyperbolic networks.

### 2.1 Graph Representation Learning

Early research on graph representations relied on learning effective node representations, primarily, through two broad methods, namely, matrix factorization and random walks. In matrix factorization based approaches Cao et al. (2015), the sparse graph adjacency matrix $A$ is factorized into low-dimensional dense matrix $L$ such that the information loss $\|L^T L - A\|$ is minimized. In the random walk based approaches Grover & Leskovec (2016); Perozzi et al. (2014); Narayanan et al. (2017), a node's neighborhood is collected with random walks through its edges, and the neighborhood is used to predict the node's representation in a dense network framework. Earlier methods such as LINE Tang et al. (2015) and SDNE Wang et al. (2016) use first-order (nodes connected by an edge) and second-order (nodes with similar neighborhood) proximity to learn the node representations. These methods form a vector space model for graphs and have shown some preliminary success. However, they are node-specific and do not consider the neighborhood information of a node or the overall graph structure. In more recent works, aggregating information from a nodes' neighborhood is explored using the neural network models. Graph neural networks (GNN) Scarselli et al. (2008), typically applied to node classification, aggregate information from a nodes' neighborhood to predict the label for the root node. Several approaches based on different neural network architectures for neighborhood aggregation have been developed in recent years and some of the popular ones include GraphSage Hamilton et al. (2017) (LSTM), Graph Convolution Networks (GCN) Kipf & Welling (2017), and Graph Attention Networks (GAT) Veličković et al. (2018). Another line of work specifically tailored for heterogeneous graphs Fu et al. (2020); Yang et al. (2021); Hu et al. (2020); Yun et al. (2019); Wang et al. (2019), utilizes the rich relational information through metapath aggregation. These approaches, while efficient at aggregating neighborhood information, *do not consider the node's semantic attributes or the global graph structure.* In the proposed TESH-GCN model, we aim to utilize the node's semantic signal, in congruence with global adjacency tensor, to capture both the node's semantic attributes and its position in the overall graph structure.

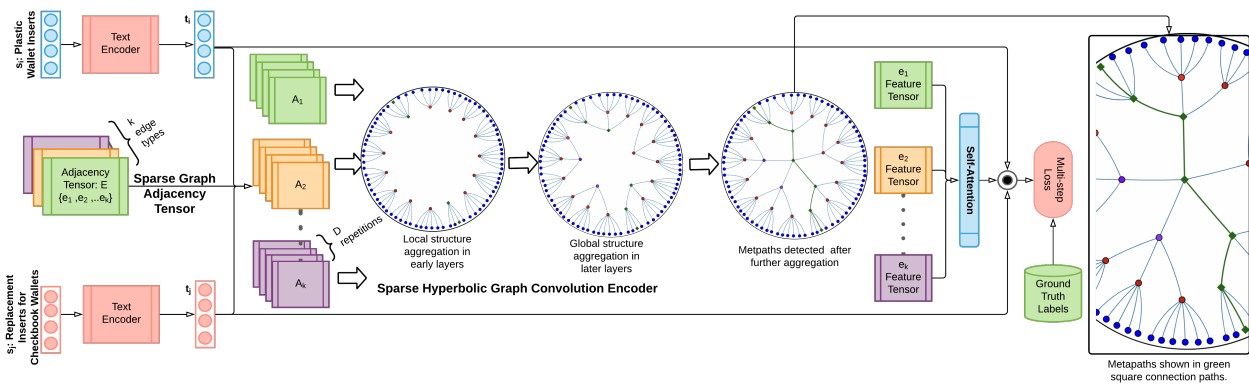

Figure 3: Architecture of our proposed model. The Hyperbolic Graph Convolution Encoder aggregates local features in the early layers and global features in the later layers. The encoder also handles sparsity to reduce both time and space complexity.

## 2.2 Hyperbolic Networks

In recent research Ganea et al. (2018), graph datasets have been shown to possess an inherent hierarchy between nodes thus demonstrating a non-Euclidean geometry. In Ganea et al. (2018), the authors provide the gyrovector space model including the hyperbolic variants of the algebraic operations required to design neural networks. The algebraic operations for the Poincaré ball of curvature $c$ are the following: Möbius addition ($\oplus_c$), exponential map ($\exp_x^c$), logarithmic map ($\log_x^c$), Möbius scalar multiplication ($\otimes_c$), and hyperbolic activation ($\sigma_c$).

$$x \oplus_c y = \frac{\left(1 + 2c\langle x, y\rangle + c\|y\|^2\right)x + \left(1 - c\|x\|^2\right)y}{1 + 2c\langle x, y\rangle + c^2\|x\|^2\|y\|^2}$$

$$\exp_x^c(v) = x \oplus_c \left(\tanh\left(\sqrt{c}\frac{\lambda_x^c\|v\|}{2}\right)\frac{v}{\sqrt{c}\|v\|}\right)$$

$$\log_x^c(y) = \frac{2}{\sqrt{c}\lambda_x^c}\tanh^{-1}\left(\sqrt{c}\| - x \oplus_c y\|\right)\frac{-x \oplus_c y}{\| - x \oplus_c y\|}$$

$$r \otimes_c x = \exp_0^c(r\log_0^c(x)), \ \forall r \in \mathbb{R}, x \in \mathbb{H}_c^n$$
$$\sigma_c(x) = \exp_0^c(\sigma(\log_0^c(x))) \tag{1}$$

where $\lambda_x^c = \frac{2}{(1-c\|x\|^2)}$ is the metric conformal factor. Based on these approaches, hyperbolic networks such as HGNN Ganea et al. (2018), HGCN Chami et al. (2019), HAN Gulcehre et al. (2019), and HypE Choudhary et al. (2021) have shown to outperform their Euclidean counterparts on graph datasets. However, these approaches still focus on the nodes' local neighborhood and not the overall graph structure. Furthermore, hyperbolic transformations are performed on entire vectors and are thus inefficient on sparse tensors. In our model, we utilize the $\beta-$split and $\beta-$concatenation operations Shimizu et al. (2021) to optimize the hyperbolic graph convolution for sparse adjacency tensors.

## 3 The Proposed model

In this section, we first describe the problem setup for link prediction on sparse heterogeneous graphs.[3] We then provide a detailed explanation of the different components of the proposed model and their functionality in the context of link prediction. The overall architecture is depicted in Figure 3. The notations used in this paper are defined in Table 1.

---

[3]Note that we use link prediction as a running example in this paper. Other tasks (node/graph classification) can be easily performed by changing the loss function.

Table 1: Notations used in the paper.

| Notation | Description |
|----------|-------------|
| $\mathcal{G}$ | the heterogeneous graph |
| $V$ | set of nodes in graph $\mathcal{G}$ |
| $K$ | number of edge types in the graph $\mathcal{G}$ |
| $E$ | $K \times |V| \times |V|$-sized boolean adjacency tensor |
| $e_k$ | $|V| \times |V|$-sized adjacency matrix edge of type $k$ in $E$ |
| $e_k(v_i, v_j)$ | boolean indicator of edge type $k$ between nodes $v_i$ and $v_j$ |
| $R$ | sparsity ratio |
| $\delta(\mathcal{G})$ | hyperbolicity of graph $\mathcal{G}$ |
| $P_\theta$ | model with parameters $\theta$ |
| $y_k$ | probability that input sample belongs to class $k$ |
| $s_i$ | textual tokens of node $v_i$ |
| $LM(x)$ | $D$-sized vector from language model $LM$ of textual tokens $x$ |
| $t_i$ | $D$-sized encoded text vector of tokens $s_i$ |
| $A_k$ | $D \times |V| \times |V|$-sized stack of adjacency matrix $e_k$ |
| $W_{f,l}$ | filter weights for feature transformation in $l^{th}$ layer |
| $o_{p,l}$ | output of feature transformation in $l^{th}$ layer |
| $\alpha_p$ | attention weights for feature aggregation in the $l^{th}$ layer |
| $a_{p,l}$ | output scaled by $\alpha_p$ in the $l^{th}$ layer |
| $h_{p,l}$ | final output of the $l^{th}$ convolution layer |
| $\alpha_k$ | attention weight of the encoding $k^{th}$ adjacency matrix |
| $h_{k,L}$ | attention scaled encoding of the $k^{th}$ adjacency matrix |
| $h_L$ | output of the sparse hyperbolic convolution layers |
| $out(A)$ | final output of TESH-GCN for input adjacency tensor A |
| $\hat{y}_k$ | ground truth labels of edge type $k$ |
| $L(y_k, \hat{y}_k)$ | cross-entropy loss over $\hat{y}_k$ and $y_k$ |

## 3.1 Problem Setup

Let us consider a heterogeneous graph $\mathcal{G} = (V, E)$ with $K$ edge types, where $v \in V$ is the set of its nodes and $e_k(v_i, v_j) \in E \in \mathbb{B}^{K \times |V| \times |V|}$ is a sparse Boolean adjacency tensor (which indicates if edge type $e_k$ exists between nodes $v_i$ and $v_j$ or not). Each node $v_i$ also contains a corresponding text sequence $s_i$. The sparsity of the adjacency tensor and hierarchy of the graph $\mathcal{G}$ is quantified by the sparsity ratio ($R$, Definition 1) and hyperbolicity ($\delta$, Definition 2), respectively. Higher sparsity ratio implies that $E$ is sparser, whereas lower hyperbolicity implies $\mathcal{G}$ has more hierarchical relations.

**Definition 1.** *Sparsity ratio (R) is defined as the ratio of the number of zero elements to the total number of elements in the adjacency tensor;*

$$R = \frac{|e_k(v_i, v_j) = 0|}{|E|} \tag{2}$$

**Definition 2.** *For a graph $\mathcal{G}$, the hyperbolicity ($\delta$) is calculated as described in Gromov (1987). Let us say $(a, b, c, d) \in \mathcal{G}$ is a set of vertices, and $dist(a, b)$ indicates the edge distance between vertices $a$ and $b$ in a homogenized version of graph $\mathcal{G}$. Let us define $S_1$, $S_2$ and $S_3$ as:*

$$S_1 = dist(a, b) + dist(d, c)$$
$$S_2 = dist(a, c) + dist(b, d)$$
$$S_3 = dist(a, d) + dist(b, c)$$

*Let $M_1$ and $M_2$ be the two largest values in $(S_1, S_2, S_3)$, then $H(a, b, c, d) = M_1 - M_2$ and $\delta(\mathcal{G})$ is given by:*

$$\delta(\mathcal{G}) = \frac{1}{2} \max_{(a,b,c,d) \in \mathcal{G}} H(a, b, c, d)$$

For the task of link prediction, given input nodes $v_i$ and $v_j$ with corresponding text sequence $s_i$ and $s_j$, respectively and an incomplete training adjacency tensor $E$, our goal is to train TESH-GCN to optimize a

predictor $P_\theta$ parameterized by $\theta$ such that;

$$y_k = P_\theta(z = 1|I)P_\theta(y = k|I), \text{ where } I = \{v_i, v_j, s_i, s_j, E\},$$

$$\theta = \arg\min_\theta \left( -\sum_{k=1}^{K} \hat{y}_k \log(y_k) \right)$$

where z is a Boolean indicator that indicates if an edge between the two nodes exists ($z = 1$) or not ($z = 0$) and y is a class predictor for each $k \in K$ edge types. $\hat{y}_k$ is the probability of each class $k \in K$ predicted by TESH-GCN and $y_k$ is the ground truth class label.

### 3.2 Text Enriched Sparse Hyperbolic GCN

In this section, we describe the message aggregation framework of TESH-GCN, which allows us to aggregate the node's text-enriched local neighborhood and long metapath features (through semantic signals and reformulated hyperbolic graph convolution) from sparse adjacency tensors in the hyperbolic space. In this section, we detail the (i) methodology of integrating semantic features with graph tensors, (ii) sparse HGCN layer to encode hierarchical and graph structure information efficiently, and (iii) aggregation through self-attention to improve model robustness.

**Incorporating Semantics into Adjacency Tensor:** To integrate the nodes' textual information with the graph structure, we need to enrich the heterogeneous graph's adjacency tensor with the nodes' semantic features. For this, we extract the nodes' semantic signals using a pre-trained language model ($LM$) Song et al. (2020). We encode the node's text sequence $s$ to a vector $t \in \mathbb{R}^D$. Each dimension of vector $t$ denotes a unique semantic feature and thus, each dimension needs to be added to a single adjacency matrix. To achieve this efficiently, let us assume that $A_k$ is a stack of D-repetitions of the adjacency matrix $e_k$. To each matrix in the stack $A_k$, we add each unique dimension of $t$ to the corresponding matrix as the nodes' semantic and positional signal particularly for that dimension (illustrated in Figure 4).

$$t_i = LM(s_i), \quad t_j = LM(s_j) \tag{3}$$

$$A_k[d, i, :] = t_i[d], \quad A_k[d, :, j] = t_j[d] \quad \forall d : 1 \to D \tag{4}$$

where $A_k[d, i, :]$ represents the $i^{th}$ row in the $d^{th}$ matrix of $A_k$ and $A_k[d, :, j]$ represents the $j^{th}$ column in the $d^{th}$ matrix of $A_k$. $t_i[d]$ and $t_j[d]$ are the $d^{th}$ dimension of their respective semantic signals. The update operations given above ensure that the adjacency tensor $A_k$ contains information on the semantic signals at the appropriate position in the graph structure. Thus, an efficient encoding of $A_k$ allows us to capture both the structural information and semantic content of the underlying nodes. We achieve this through the sparse HGCN layer.

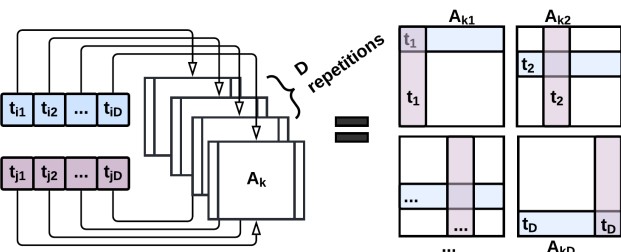

Figure 4: Adding semantic signals to the sparse adjacency tensor. The addition focuses the convolution on the highlighted areas (due to the presence of non-zeros) to initiate the extraction of graph features at the location of the input nodes.

**Sparse Hyperbolic Graph Convolution:** To encode the graph structure and latent hierarchy, we need to leverage the adjacency tensor's sparsity in the hyperbolic space for computational efficiency. To achieve

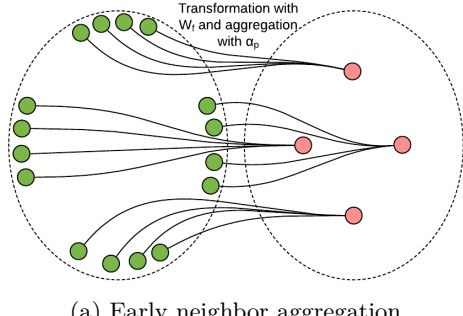
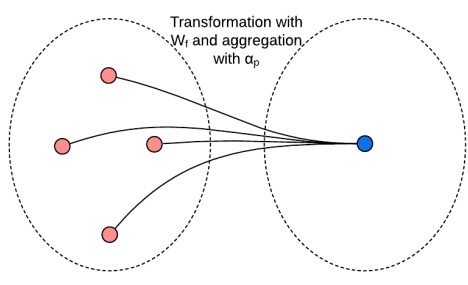

(a) Early neighbor aggregation          (b) Later metapath aggregation

Figure 5: Interpretation of the hyperbolic graph convolution. The first few layers aggregate neighborhood information and the later layers aggregate graph-level metapath information. Darker cells indicate higher weight values.

this, we reformulate the hyperbolic graph convolution in the following manner. The graph convolution layer has two operations, namely, feature transformation and aggregation, which are achieved through convolution with a filter map of trainable curvature and pooling, respectively. For a matrix of size $m_r \times m_c$ and filter map $f \times f$, graph convolution requires $\approx (m_r - f) \times (m_c - f)$ operations. However, given the high sparsity of adjacency matrices, operations on zero-valued cells will return zero gradients and, thus not contribute to the learning process. Hence, we only apply the filter transformation to adjacency tensor cells with nonzero values and ignore the zero-valued cells. For the $d^{th}$ input adjacency matrix with elements $x \in A_k[d]$,

$$o_{p,l} = W_{f,l} \otimes_{c_l} x_{p,l-1} \oplus_{c_l} b_l \quad \forall x_{p,l-1} \neq 0 \tag{5}$$

$$a_{p,l} = exp_{x_{p,l-1}}^{c_l} \left( \frac{\alpha_p \log_{x_{p,l-1}}^{c_l}(o_{p,l})}{\sum_p \alpha_p \log_{x_{p,l-1}}^{c_l}(o_{p,l})} \right) \tag{6}$$

$$h_{p,l} = \sigma_{c_l}(a_{p,l}) \tag{7}$$

where $o_{p,l}$ represents the output of feature transformation at the layer $l$ for non-zero input elements $x_{p,l-1}$ of previous layer's $l-1$ adjacency tensor with learnable feature map $W_{f,l}$. $c_l$ and $b_l$ represent the Poincaré ball's curvature and bias at layer $l$, respectively. $\otimes_{c_l}$ and $\oplus_{c_l}$ are the Möbius operations of addition and scalar multiplication, respectively. $a_{p,l}$ is the output of the scalar-attention Vaswani et al. (2017) over the outputs with attention weights $\alpha_p$ and $h_{p,l}$ is the layer's output after non-linear hyperbolic activation. The initial layers aggregate the sparse neighborhoods into denser cells. As the adjacency tensors progress through the layers, the features are always of a lower resolution than the previous layer (aggregation over aggregation), and thus aggregation in the later layers results in graph-level metapath features, as depicted in Figure 5. Note that the computational complexity of calculating $o_{p,l}$ in sparse graph convolutions is $\mathcal{O}(V^2(1 - R))$ when compared to $\mathcal{O}(V^2)$ of dense graph convolutions[4]. This indicates a reduction in the total number of computations by a factor of $(1-R) \approx 10^{-4}$. Prior hyperbolic approaches could not utilize sparse convolutions because the hyperbolic operation could not be performed on splits of the adjacency tensor but we enable this optimization in TESH-GCN through the operations of $\beta$-split and $\beta$-concatenation Shimizu et al. (2021), formulated in Definition 3 and 4.

Let us say, the $d$-dimensional hyperbolic vector in Poincaré ball of curvature $c$ is $x \in \mathbb{H}_c^d$ and $\beta_d = B\left(\frac{d}{2}, \frac{1}{2}\right)$ is a scalar beta coefficient, where B is the beta function. Then, the *$\beta$-split* and *$\beta$-concatenation* are defined as follows.

**Definition 3.** *$\beta$-split: The hyperbolic vector is split in the tangent space with integer length $d_i : \Sigma_{i=1}^D d_i = d$ as $x \mapsto v = log_0^c(x) = (v_1 \in \mathbb{R}^{d_1}, ..., v_D \in \mathbb{R}^{d_D})$. Post-operation, the vectors are transformed back to the hyperbolic space as $v \mapsto y_i = exp^c(\beta_{d_i} \beta_d^{-1} v_i)$.*

---

[4]Practically, for a graph with $10^5$ nodes and a sparsity of $10^{-4}$, dense graph convolution requires 80GB of memory (assuming double precision) for one layer, whereas, sparse graph convolution only requires 1MB of memory for the same. This allows us to utilize the entire adjacency tensor, while previous approaches can only rely on the local neighborhood.

**Definition 4.** *β-concatenation: The hyperbolic vectors to be concatenated are transformed to the tangent space, concatenated and scaled back using the beta coefficients as;* $x_i \mapsto v_i = log_0^c(x_i), v := (\beta_d \beta_{d_1}^{-1} v_1, ..., \beta_d \beta_{d_D}^{-1} v_D) \mapsto y = exp^c(v).$

The final encoding of an adjacency tensor $A_k$ is, thus, the output features of the last convolution layer transformed to the tangent space with the logarithmic map $h_{k,L} = log_0^{c_L}(h_{k,L})^5$.

**Aggregation through Self-Attention:** Given the encoding of adjacency tensor of all edge types $A_k \in A$, we aggregate the adjacency tensors such that we capture their inter-edge type relations and also condition our prediction on both the graph and text for robustness. To achieve this, we pass the adjacency tensor encodings $A_k \in A$ through a layer of self-attention Vaswani et al. (2017) to capture the inter- edge type relations through attention weights. The final encoder output $out(A)$ concatenates the features of adjacency tensor with the semantic embeddings to add conditionality on both graph and text information.

$$h_{k,L} = \frac{\alpha_k h_{k,L}}{\sum_k \alpha_k h_{k,L}} \tag{8}$$

$$h_L = h_{1,L} \odot h_{2,L} \odot \cdots \odot h_{k,L} \tag{9}$$

$$out(A) = h_L \odot t_i \odot t_j \tag{10}$$

where $\alpha_k$ are the attention weights of edge types and $h_L$ are the adjacency tensors' features. The semantic residual network connection sends node signals to the adjacency tensor and also passes information to the multi-step loss function. The balance between semantic residual network and hyperbolic graph convolution leads to robustness against noisy text or graphs (evaluated empirically in Section 4.6).

## 3.3 Multi-step Loss

In this work, we consider a generalized link prediction problem in heterogeneous networks where there are two sub-tasks. (i) To predict if a link exists between two nodes and (ii) To predict the class/type of link (if one exists). One method to achieve this goal is to add the non-existence of link as another class. Let us assume we add a class $z$ which indicates the existence of the link ($z = 1$) and $z = 0$ when the link is absent. Then, for the task of link prediction, we need to support the independence assumption, i.e., $z \perp\!\!\!\perp e_k, \quad \forall e_k \in E$, which is not true. Prediction of an edge type $e_k$ is conditional on $z = 1$. Hence, we setup a multi-step loss that first predicts the existence of a link and then classifies it into an edge type.

$$y_k = P_\theta(e_k|x) = P_\theta(z = 1|x)P_\theta(y = e_k|x) \tag{11}$$

$$L(y_k, \hat{y_k}) = -\sum_{k=1}^{K} \hat{y_k} \log(y_k) \tag{12}$$

where $x$ and $\theta$ are the input and model parameters, respectively. $L$ is the cross entropy loss that needs to be minimized. Although we use this generalized link prediction as the task of interest in this paper, TESH-GCN can be applied to any task such as node/graph classification by replacing the loss with the appropriate loss.

## 3.4 Implementation Details

We implemented TESH-GCN using Pytorch Paszke et al. (2019) on eight NVIDIA V100 GPUs with 16 GB VRAM. For gradient descent, we used Riemmanian Adam Becigneul & Ganea (2019) with standard $\beta$ values of 0.9 and 0.999 and an initial learning rate of 0.001. Number of dimensions ($D$) and number of layers ($L$) is empirically selected based on performance-memory trade-off. Figure 6 presents the memory-performance trade-off for different choices of parameters D and L. We observe that the $D = 8$ and $L = 8$ provides the best performance for the memory required. Hence, we chose them for the final implementation of our

---

[5]The transformation from hyperbolic space to tangent space with logarithmic map is required for attention-based aggregation as such formulation is not well-defined for the hyperbolic space.

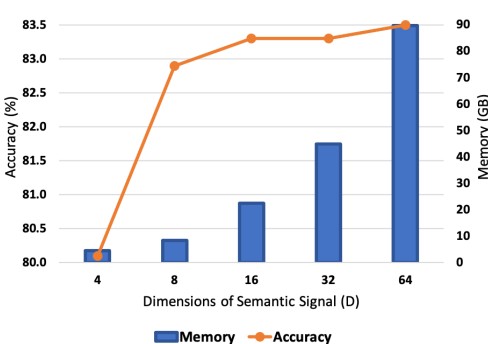 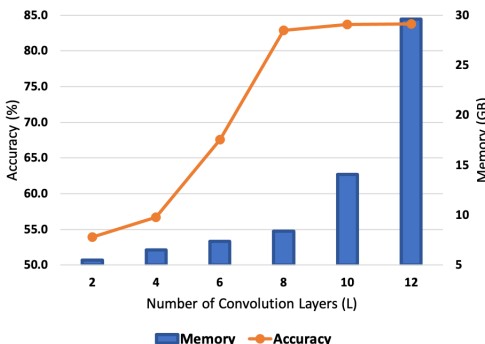

(a) Dimension of semantic signal (D) vs Memory and Accuracy.

(b) No. of graph convolution layers (L) vs Memory and Accuracy.

Figure 6: Effect of L and D parameters on memory required and accuracy performance of TESH-GCN on Amazon dataset. Note that we use 16GB of Nvidia V100 GPU for our experiments. For higher than 16GB of memory we place different components on different GPU and moving the tensors among different GPUs adds an insignificant overhead.

---

**Algorithm 1:** TESH-GCN training

**Data:** Training data $(v_i, s_i, v_j, s_j, \hat{y}_k) \in E$;
**Output:** Predictor $P_\theta$;

**1** Initialize model parameters $\theta$;
**2** **for** *number of epochs; until convergence* **do**
**3**   $l = 0$; # Initialize loss
**4**   **for** $\{(v_i, s_i, v_j, s_j, \hat{y}_k) \in E\}$ **do**
**5**     $t_i \leftarrow LM(s_i)$, $t_j \leftarrow LM(s_j)$;
**6**     **for** $e_k \in E$ **do**
**7**       # Stack D-repetitions of adjacency matrix
**8**       $A_k = stack(E_k, D)$;
**9**       $A_k[d, i, :] = t_i[d]$, $A_k[d, j, :] = t_j[d]$
**10**      $x_0 = A_k$
**11**      # Run through $L$ graph convolution layers
**12**      **for** $l : 1 \rightarrow L$ **do**
**13**        $o_{p,l} = W^f \otimes_{c_l} x_{p,l-1} \oplus_{c_l} b_l \quad \forall x_{p,l-1} \neq 0$
**14**        $a_{p,l} = exp^{c_l}\left(\frac{\alpha_p \log^{c_l}(o_{p,l})}{\sum_p \alpha_p \log^{c_l}(o_{p,l})}\right)$
**15**        $h_{p,l} = \sigma_{c_l}(a_{p,l})$
**16**      **end**
**17**      $h_{k,L} = h_{p,l}$
**18**    **end**
**19**    # Attention over outputs
**20**    $h_{k,L} = \frac{\alpha_k h_{k,L}}{\sum_k \alpha_k h_{k,L}}$
**21**    $h_L = h_{1,L} \odot h_{2,L} \odot ... \odot h_{k,L}$
**22**    $out(A) = h_L \odot t_i \odot t_j$
**23**    # Predicted class probability
**24**    $y_k = softmax(dense(out(A)))$
**25**    $l = l + L(y_k, \hat{y}_k)$  # Update loss
**26**  **end**
**27**  $\theta \leftarrow \theta - \nabla_\theta l$;  # Update parameters
**28** **end**
**29** **return** $P_\theta$

model. For non-linearity, we used the hyperbolic activation function, given in Eq. (1). The sparsity in the model variables is handled using the torch-sparse library[6]. While this library and other similar ones handle the operational sparsity of the graphs, previous GNN-based approaches need to locally convert the sparse tensors to the corresponding dense format for their layer operations. In TESH-GCN, the conversion is not required because **all operations in Sparse-HGCN are directly performed on the sparse tensor as it only considers the non-zero elements of the tensor.** Each convolution operation moves up one-hop in the nodes' neighborhood. Hence, the number of graph convolution layers should at least be the maximum shortest path between any two nodes in the graph. For a dataset, this is empirically calculated by sampling nodes from the graph and calculating the maximum shortest path between them. For the datasets in our experiments, we used 8 layers ($L = 8$) to extract local neighborhoods in the early layers and metapath structures in the later layers. The main adjacency tensor can be split either over the number of semantic signals (D) or the number of edge types (K). We chose the latter because each adjacency tensor needed a separate GPU and it was more efficient and convenient to control the training process, given that the number of edge types is lesser than the number of semantic signals in our experiments. Algorithm 1 provides the pseudocode for training the model.

## 4  Experimental Setup

In this section, we describe our experimental setup and investigate the following research questions (**RQs**):

1. **RQ1:** Does TESH-GCN perform better than the state-of-the-art approaches for the task of link prediction?
2. **RQ2:** What is the contribution of TESH-GCN's individual components to the overall performance?
3. **RQ3:** How does TESH-GCN compare against previous approaches in time and space complexity?
4. **RQ4:** How robust is TESH-GCN against noise in the graph and its corresponding text?
5. **RQ5:** Can we comprehend the results of TESH-GCN?

Table 2: Dataset statistics including no. of nodes (V), edges (E), edge types (K), hyperbolicity ($\delta$), and sparsity ratio (R).

| Dataset | V | E | K | $\delta$ | R (%) |
|---|---|---|---|---|---|
| Amazon | 368,871 | 6,471,233 | 2 | 2 | 99.99 |
| DBLP | 37,791 | 170,794 | 3 | 4 | 99.99 |
| Twitter | 81,306 | 1,768,149 | 1 | 1 | 99.97 |
| Cora | 2,708 | 5,429 | 1 | 11 | 99.92 |
| MovieLens | 10,010 | 1,122,457 | 3 | 2 | 99.00 |

### 4.1  Datasets Used

For the datasets, we select the following widely used publicly available network benchmark datasets where the nodes contain certain semantic information in the form of text attributes. Also, the choice of the datasets is driven by the diversity of their hyperbolicity to test performance on different levels of latent hierarchy (lower hyperbolicity implies more latent hierarchy).

1. **Amazon** He & McAuley (2016) is a heterogeneous e-commerce graph dataset that contains electronic products as nodes with title text connected by edges based on the purchase information. The edge types are `also_buy` (products bought together) and `also_view` (products viewed in the same user session).
2. **DBLP** Ji et al. (2010) is a heterogeneous relational dataset that contains papers, authors, conferences, and terms from the DBLP bibliography website connected by three edge types: `paper-author`, `paper-conf` and `paper-term`. For the semantic information, we include the paper's titles, author's names, conference's names, and the terms' text.

---

[6]https://github.com/rusty1s/pytorch_sparse

3. **Twitter** Leskovec & Mcauley (2012) dataset is a user follower network graph with unidentifiable profile information given as node's features. The node features are pre-encoded to remove sensitive identifiable information.
4. **Cora** Rossi & Ahmed (2015) is a citation graph that contains publications with title text and author information connected by citation edges.
5. **MovieLens** Harper & Konstan (2015) dataset is a standard user-movie heterogeneous rating dataset with three edge types: `user-movie`, `user-user`, and `movie-genre`. We utilize the movie's title and genre's name as the textual information.

In the case of graph-based methods, we utilize the node features provided in the dataset as default, else we utilize fixed-semantic vectors from the pretrained LM Song et al. (2020). More detailed dataset statistics such as the number of nodes, edges, edge types, along with hyperbolicity and sparsity are given in Table 2.

## 4.2 Baselines

We compare the performance of the proposed model with the following state-of-the-art models in the following categories: text-based (1-3), graph-based (4-6), and hybrid text-graph (7-9) approaches.

1. **C-DSSM** Shen et al. (2014) is an extension of DSSM Huang et al. (2013) that utilizes convolution layers to encode character trigrams of documents for matching semantic features.

2. **BERT** Devlin et al. (2019) is a popular transformer based language model that pre-trains on large amount of text data and is fine-tuned on sequence classification task for efficient text matching.

3. **XLNet** Yang et al. (2019) is an improvement over the BERT model which uses position invariant autoregressive training to pre-train the language model.

4. **GraphSage** Hamilton et al. (2017) is one of the first approaches that aggregate the neighborhood information of a graph's node. It includes three aggregators mean, LSTM Hochreiter & Schmidhuber (1997), and max pooling. For our baseline, we choose the best performing LSTM aggregator.

5. **GCN** Kipf & Welling (2017) utilizes convolutional networks to aggregate neighborhood information.

6. **HGCN** Chami et al. (2019) utilizes convolutional networks in the hyperbolic space that typically performs better than the Euclidean counterparts, especially, for datasets with low hyperbolicity (i.e., more latent hierarchy).

7. **TextGNN** Zhu et al. (2021) initializes node attributes with semantic embeddings to outperform previous approaches especially for the task of link prediction.

8. **TextGCN** Yao et al. (2019) constructs a word-document graph based on TF-IDF scores and then applies graph convolution for feature detection towards link prediction between nodes.

9. **Graphormer** Ying et al. (2021) adds manually constructed global features using spatial encoding, centrality encoding, and edge encoding to the node vector to aggregate the neighborhood in a transformer network architecture for graph-level prediction tasks.

## 4.3 RQ1: Performance on Link Prediction

To analyze the performance of TESH-GCN, we compare it against the state-of-the-art baselines using standard graph datasets on the task of link prediction. We input the node-pairs $(v_i, v_j)$ with the corresponding text sequence $(s_i, s_j)$ to the model and predict the probability that an edge type $e_k$ connects them as $y_k = P_\theta(e_k|(v_i, v_j, s_i, s_j))$. We evaluate our model using 5-fold cross validation splits on the following standard performance metrics: Accuracy (ACC), Area under ROC curve (AUC), Precision (P), and F-score (F1). For our experimentation, we perform 5-fold cross validation with a training, validation and test split of 8:1:1 on the edges of the datasets. Table 3 provides the number of samples and sparsity of each split in the dataset. The results on the test set are presented in Table 4.

Table 3: Splits of the dataset for the link prediction experiment (RQ1). N is the number of samples in each split and R(%) provides the sparsity ratio of the split.

| Dataset | Training | | Validation | | Test | |
|---|---|---|---|---|---|---|
| | N | R(%) | N | R(%) | N | R(%) |
| **Amazon** | 5,176,986 | 99.99 | 647,123 | 99.99 | 647,124 | 99.99 |
| **DBLP** | 1,36,635 | 99.99 | 17,079 | 99.99 | 17,080 | 99.99 |
| **Twitter** | 1,414,519 | 99.97 | 176,815 | 99.99 | 176,815 | 99.99 |
| **Cora** | 4,343 | 99.94 | 543 | 99.99 | 543 | 99.99 |
| **MovieLens** | 897,966 | 99.10 | 112,245 | 99.88 | 112,246 | 99.88 |

From the experimental results, we observe that TESH-GCN is able to outperform the previous approaches by a significant margin on different evaluation metrics. Additionally, we notice that the performance improvement of hyperbolic models (HGCN and TESH-GCN) is more on datasets with lower hyperbolicity (higher latent hierarchy). This shows that hyperbolic space is better at extracting hierarchical features from the graph structures. Furthermore, we see that the performance decreases a little without the residual network. However, it does not justify the additional parameters but it adds robustness against noisy graph and text (evaluation in Section 4.6), so we use this variant in our final model. Another point of note is that text-based frameworks are better than graph approaches in datasets with good semantic information such as Amazon, whereas, graph-based approaches are better on well-connected graphs such as Cora. However, TESH-GCN is able to maintain good performance in both the scenarios, *demonstrating its ability to capture both semantic and structural information from the dataset.*

Table 4: Performance comparison of our proposed model against several state-of-the-art baseline methods across diverse datasets on the task of link prediction. Metrics such as Accuracy (ACC), Area under ROC (AUC), Precision (P), and F-scores (F1) are used for evaluation. The rows corresponding to w/o Text, w/o Hyperbolic, w/o Residual, and CE Loss represent the performance of TESH-GCN without the text information, hyperbolic transformation, residual connections, and with standard cross entropy loss (instead of multi-step loss), respectively. The best and second best results are highlighted in bold and underline, respectively. The improvement of TESH-GCN is statistically significant over the best performing baseline with a p-value threshold of 0.01.

| Datasets Models | | Amazon | | | | DBLP | | | | Twitter | | | | Cora | | | | MovieLens | | | |
|---|---|---|---|---|---|---|---|---|---|---|---|---|---|---|---|---|---|---|---|---|---|---|
| | | ACC | AUC | P | F1 | ACC | AUC | P | F1 | ACC | AUC | P | F1 | ACC | AUC | P | F1 | ACC | AUC | P | F1 |
| Text | C-DSSM | .675 | .681 | .677 | .674 | .518 | .522 | .519 | .513 | .593 | .595 | .588 | .586 | .693 | .697 | .696 | .693 | .664 | .660 | .658 | .660 |
| | BERT | .787 | .793 | .797 | .784 | .604 | .605 | .605 | .603 | .667 | .664 | .630 | .641 | .757 | .763 | .758 | .751 | .760 | .764 | .757 | .752 |
| | XLNet | .788 | .793 | .797 | .785 | .602 | .602 | .610 | .604 | .626 | .626 | .651 | .654 | .761 | .768 | .762 | .758 | .750 | .758 | .766 | .754 |
| Graph | GraphSage | .677 | .680 | .679 | .673 | .520 | .525 | .519 | .518 | .591 | .592 | .588 | .585 | .809 | .813 | .813 | .805 | .660 | .659 | .662 | .656 |
| | GCN | .678 | .679 | .679 | .674 | .412 | .412 | .413 | .401 | .564 | .566 | .553 | .545 | .813 | .817 | .818 | .814 | .652 | .652 | .649 | .650 |
| | HGCN | .710 | .715 | .712 | .703 | .547 | .548 | .544 | .533 | .608 | .605 | .580 | .598 | **.929** | **.934** | **.931** | **.923** | .685 | .697 | .687 | .677 |
| Hybrid | TextGNN | .742 | .742 | .744 | .732 | .567 | .573 | .573 | .562 | ..636 | .636 | .628 | .621 | .843 | .848 | .848 | .840 | .723 | .724 | .719 | .712 |
| | TextGCN | .817 | .824 | .818 | .809 | .624 | .626 | .625 | .616 | .671 | .670 | .660 | .669 | .862 | .864 | .870 | .856 | .789 | .790 | .783 | .780 |
| | Graphormer | .804 | .808 | .806 | .804 | .617 | .619 | .621 | .612 | .692 | .693 | .669 | .666 | .849 | .851 | .858 | .849 | .780 | .780 | .779 | .771 |
| Ours | TESH-GCN | **.829** | **.836** | **.837** | **.836** | **.636** | **.640** | **.644** | **.640** | **.709** | **.710** | **.671** | **.670** | .909 | .901 | .902 | .908 | **.806** | **.814** | **.801** | **.801** |
| | w/o Text | .784 | .784 | .784 | .784 | .599 | .605 | .612 | .599 | .645 | .648 | .648 | .622 | .854 | .858 | .842 | .824 | .759 | .753 | .756 | .748 |
| | w/o Hyperbolic | .677 | .672 | .678 | .678 | .522 | .526 | .531 | .516 | .577 | .572 | .554 | .585 | .787 | .789 | .781 | .757 | .655 | .652 | .651 | .660 |
| | w/o Residual | .826 | .825 | .829 | .829 | .629 | .632 | .640 | .632 | .699 | .705 | .662 | .658 | .937 | .942 | .929 | .913 | .796 | .799 | .788 | .795 |
| | CE Loss | .827 | .830 | .833 | .832 | .635 | .635 | .642 | .639 | .706 | .707 | .668 | .665 | .931 | .939 | .927 | .916 | .800 | .805 | .798 | .795 |

## 4.4 RQ2: Ablation Study

In this section, we study the importance of different components and their contribution to the overall performance of our model. The different components we analyze in our ablation study are: (i) the semantic text signal, (ii) the hyperbolic transformations, (iii) the residual network, and (iv) the multi-step loss. The ablation study is conducted on the same datasets by calculating the evaluation metrics after freezing the parameters of the component of interest in the model. The results of the study are presented in Table 4.

The results show that the text signal contributes to 7% performance gain in our model, implying the importance of utilizing the nodes' semantic information in aggregating features from the adjacency tensors. The hyperbolic transformations lead to a 18% increase in TESH-GCN's performance, demonstrating the importance of hierarchical features in extracting information from graphs. This also provides additional evidence of the latent hierarchy in the graph networks. Furthermore, removing the residual network shows a decrease of 1% in our model's performance which shows that text signals capture the semantic signal in the graph convolution layers and the residual network works only towards increasing the robustness in the final link prediction task. In addition to this, we notice that replacing multi-step loss with a standard cross entropy loss (with non-existence of links added as another class) leads to a 2% reduction in performance. This provides evidence for the advantages of conditioning link classification on link prediction (as in multi-step loss) compared to a standard multi-class loss function.

### 4.5 RQ3: Complexity Analysis

*One of the major contributions of TESH-GCN is its ability to efficiently handle sparse adjacency tensors in its graph convolution operations.* To compare its performance to previous graph-based and hybrid approaches, we analyze the space and time complexity of our models and the baselines. The space complexity is studied through the number of parameters and time complexity is reported using the training and inference times of the models. We compare the space and time complexity of our models using large graphs of different sparsity ratios ($R$) (by varying the number of edges/links on a graph with $10^4$ nodes). The different sparsity ratios considered in the evaluation are $1 - 10^{-r} \ \forall r \in [\![0, 5]\!]$. Figure 7 and Table 5 shows the comparison of different GCN based models' training time on varying sparsity ratios and inference times on different datasets, respectively. Table 6 presents the number of parameters and space complexity of the different baselines in comparison to TESH-GCN. From the time complexity analysis, we notice that TESH-GCN

Table 5: Inference times (in milliseconds) of our model and various GCN-based baseline methods.

| Models | Amazon | DBLP | Twitter | Cora | MovieLens |
|---|---|---|---|---|---|
| GCN | 719 | 723 | 728 | 735 | 744 |
| HGCN | 745 | 757 | 758 | 763 | 774 |
| TextGNN | 1350 | 1368 | 1375 | 1394 | 1395 |
| TextGCN | 1392 | 1416 | 1417 | 1431 | 1437 |
| Graphormer | 1423 | 1430 | 1441 | 1442 | 1458 |
| TESH-GCN | 787 | 794 | 803 | 817 | 822 |

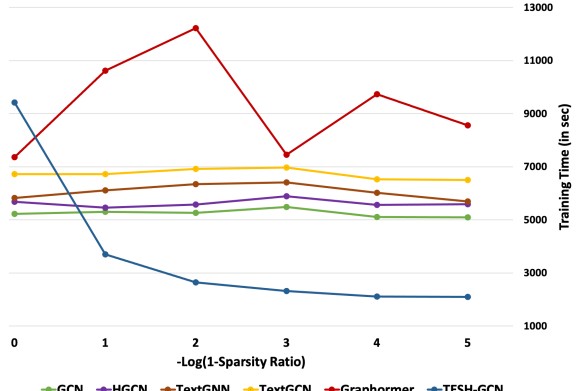

Figure 7: Comparison of training time (in seconds) of different GCN-based baseline methods on datasets with varying sparsity ratios (R).

consistently takes much less training time than the other GCN-based and hybrid approaches in high sparsity graphs. This shows that the current GCN-based approaches do not handle the sparsity of the adjacency tensor. However, the overhead of specialized graph convolution layer in TESH-GCN leads to a poor time complexity for cases with high graph density ($R < 0.9$). From the comparison of inference times, given in Table 5, we notice that TESH-GCN's inference time is comparable to the graph-based baselines and significantly lesser than hybrid baselines. Figure 8 provides the effect of sparsity on the inference time of our model and the baselines. We note that TESH-GCN is able to outperform other hybrid graph-text baselines

Table 6: The number of non-trainable (in millions) and trainable (in thousands) parameters of all the comparison methods. We also report the space complexity in terms of the number of nodes (V), maximum text length (S), and sparsity measure $\left( N = \frac{1}{1-R} \approx 10^4 \right)$.

| Model | Non-Train (M) | Train (K) | Complexity |
|---|---|---|---|
| C-DSSM | 0 | 38 | $\mathcal{O}(S)$ |
| BERT | 110 | 1600 | $\mathcal{O}(S^2)$ |
| XLNet | 110 | 1600 | $\mathcal{O}(S^2)$ |
| GraphSage | 0 | 4800 | $\mathcal{O}(V^2)$ |
| GCN | 0 | 4800 | $\mathcal{O}(V^2)$ |
| HGCN | 0 | 9600 | $\mathcal{O}(2V^2)$ |
| TextGNN | 110 | 6400 | $\mathcal{O}(SV^2)$ |
| TextGCN | 110 | 6400 | $\mathcal{O}(SV^2)$ |
| Graphormer | 100 | 7600 | $\mathcal{O}(SV^2)$ |
| TESH-GCN | 110 | 78 | $\mathcal{O}\left( \frac{2SV^2}{N} \right)$ |

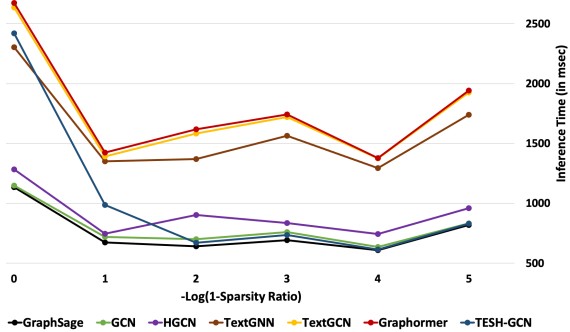

Figure 8: -log(1-R) vs Inference time (in milliseconds). Comparison of inference time of different baselines on a simulated dataset with 10,000 nodes and varying sparsity ratios (R).

and needs similar inference time as the baselines that only consider the local neighborhood of its nodes. TESH-GCN is faster for high sparsity graphs but the overhead of specialized graph convolutions takes more time than other baselines on high density graphs.

The space complexity analysis clearly shows that TESH-GCN uses much lesser number of model parameters than baselines with comparable performance. Also, the complexity shows the dependence of text-based approaches on only the textual sequence length, whereas, the graph based are dependent on the number of nodes. However, TESH-GCN is able to reduce the space complexity by a factor of the sparsity ratio and only consider informative non-zero features from the adjacency tensors, leading to a decrease in the number of trainable parameters.

### 4.6 RQ4: Model Robustness

To test the robustness of our model, we introduce varying levels of noise into the Amazon graph by (i) *node drop*: dropping n% percentage of nodes, (ii) *text replacement*: replacing n% percentage of the text, and (iii) *hybrid noise*: dropping n% of nodes and replacing n% of text. We compare the performance of our model and the baselines across different values of $n = 10, 20, 30, 40,$ and 50. The results for the robustness evaluation are given in Figure 9.

First, we highlight the main observations, that node drop and text replacement only affects graph-based and text-based approaches, respectively (and does not affect them vice versa). In the case of hybrid baselines, we still note a decrease in performance for both the noise variants. This implies that the text and graph features in the baselines do not complement each other. In the case of TESH-GCN, we note that both the noise variants do not cause any significant performance loss. This shows that the complementary nature of the semantic residual network and hyperbolic graph convolution network leads to an increased robustness against noise in either the text or graph. In the third scenario with hybrid noise, we see a reduction of $\approx 25\%$ performance in text-based and graph-based baselines and $\approx 50\%$ in hybrid baseline with a 50% noise.

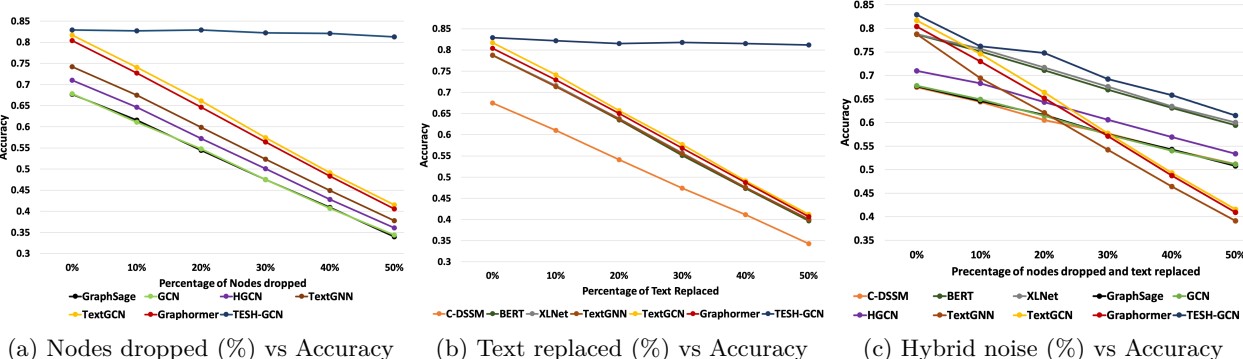

(a) Nodes dropped (%) vs Accuracy    (b) Text replaced (%) vs Accuracy    (c) Hybrid noise (%) vs Accuracy

Figure 9: Comparison of the effect of different noise-inducing methods on the accuracy of our model and the baselines. Noise is induced using (a) Node drop, (b) Text replacement, and (c) Hybrid noise (node drop and text replacement).

However, we notice that, although TESH-GCN is a hybrid model, we only observe a 25% performance loss with 50% noise, implying the effectiveness of text-graph correspondence in the scenario of hybrid noise as well. Thus, we conclude that TESH-GCN is robust against noise in either graph or text, but vulnerable, albeit less than other hybrid baselines, to a joint attack on both the graph and text.

---

**Algorithm 2:** Explaining results through Metapaths

**Input:** Input $(v_i, s_i, v_j, s_j)$, Predictor $P_\theta$;
**Output:** Metapath set $M$, Class prediction $y_k$;

**1** Initialize metapath set $M = \phi$;
**2** $t_i \leftarrow LM(s_i)$, $t_j \leftarrow LM(s_j)$;
**3 for** $e_k \in E$ **do**
**4**      Initialize metapath for $e_k$, $M_k = \phi$;
**5**      # stack D-repetitions of adjacency matrix
**6**      $A_k = stack(E_k, D)$;
**7**      $A_k[d, i, :] = t_i[d]$, $A_k[d, j, :] = t_j[d]$
**8**      $x_0 = A_k$
**9**      # Run through $L$ graph convolution layers
**10**      **for** $l : 1 \rightarrow L$ **do**
**11**          $o_{p,l} = W^f \otimes_{c_l} x_{p,l-1} \oplus_{c_l} b_l \quad \forall x_{p,l-1} \neq 0$
**12**          $a_{p,l} = exp^{c_l}\left(\frac{\alpha_p \log^{c_l}(o_{p,l})}{\sum_p \alpha_p \log^{c_l}(o_{p,l})}\right)$
**13**          $h_{p,l} = \sigma_{c_l}(a_{p,l})$
**14**          $M_k = M_k \cup \arg\max_p h_{p,l}$
**15**      **end**
**16**      $h_{k,L} = h_{p,l}$
**17 end**
**18** # Attention over outputs
**19** $h_{k,L} = \frac{\alpha_k h_{k,L}}{\sum_k \alpha_k h_{k,L}}$
**20** # Extracted metapath $M_k$ with attention weight $\alpha_k$
**21** $M = M \cup (M_k, \alpha_k)$
**22** $h_L = h_{1,L} \odot h_{2,L} \odot ... \odot h_{k,L}$
**23** $out(A) = h_L \odot t_i \odot t_j$
**24** # Predicted class probability
**25** $y_k = softmax(dense(out(A)))$
**26 return** $M, y_k$

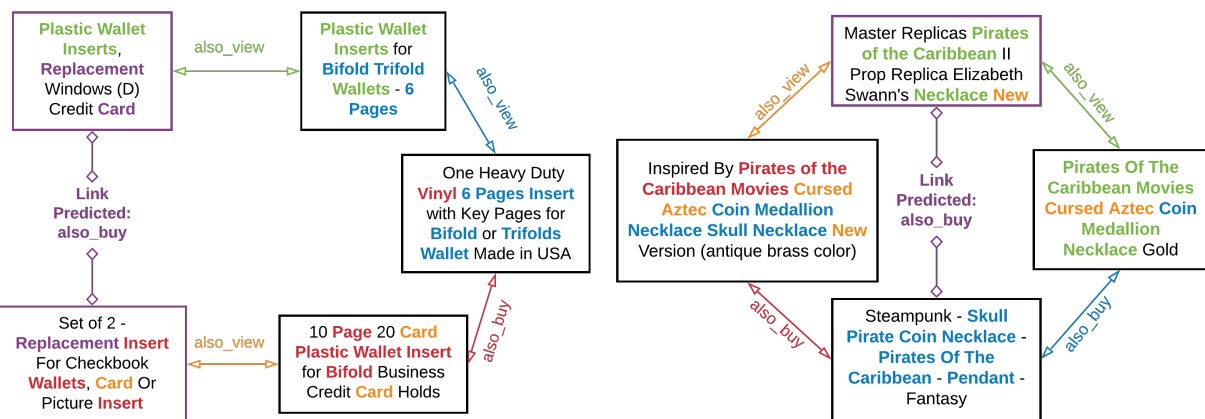

(a) Aggregating information from long metapaths.    (b) Aggregating information from multiple metapaths.

Figure 10: Predictions showing TESH-GCN's metapath aggregation ability over both text and graphs. The local neighborhood and long metapath information is extracted in the early and later graph convolution layers, respectively. The textual information is extracted using attention over the semantic residual network. The colors assigned to the text match the color of the link through which the semantic information was passed to the ultimate nodes for message aggregation and subsequently link prediction. The samples are taken from the heterogeneous Amazon dataset.

### 4.7 RQ5: Model Explainability

Model comprehension is a critical part of our architecture as it helps us form a better understanding of the results and explain the model's final output. To understand TESH-GCN's link prediction, we look at the different metapaths that connect the input nodes as well as the text in the metapaths' nodes that receive the most attention ($\alpha_k$). For this, we follow the graph convolution and attention pooling operations through the layers in the network and extract the most critical metapaths chosen by the model to arrive at the prediction. The methodology for extracting the metapaths with their corresponding weightage in the final link prediction is presented in Algorithm 2. Figure 10 depicts some metapaths extracted from the Amazon dataset. In Figures 10a and 10b, we note that TESH-GCN aggregates information from multiple long (4-hop) metapaths between the input nodes for prediction. Additionally, we see tokens in the node's text being emphasized (having higher attention weight) based on the edges through which they propagate their semantic information, e.g., in Figure 10b, we observe that key tokens: `Pirates of the Caribbean` and `Necklace` propagate the semantic information to match with additional relevant tokens such as `Cursed Aztec`, `Medallion`, `Pendant` and `coin` to establish the edge `also_buy` between the input nodes. Thus, *we observe the role of different metapaths as well as semantic information in the message propagation towards the downstream task of link prediction.*

## 5 Conclusion

In this paper, we introduced Text Enriched Sparse Hyperbolic Graph Convolution Network (TESH-GCN), a hybrid graph and text based model for link prediction. TESH-GCN utilizes semantic signals from nodes to aggregate intra-node and inter-node information from the sparse adjacency tensor using a reformulated hyperbolic graph convolution layer. We show the effectiveness of our model against the state-of-the-art baselines on diverse datasets for the task of link prediction and evaluate the contribution of its different components to the overall performance. Additionally, we demonstrate the optimized memory and faster processing time of our model through space and time complexity analysis, respectively. Furthermore, we also show TESH-GCN's robustness against noisy graphs and text and provide a mechanism for explaining the results produced by the model.

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
