# OpenReview forum: "TESH-GCN: Text Enriched Sparse Hyperbolic Graph Convolutional Networks"
_TMLR — Rejected by TMLR_

### Review · Reviewer_t96k · 2022-11-27

**Summary Of Contributions:**

This paper proposes TESH-GCN, a hyperbolic graph neural network for e-commerce link prediction on heterogeneous graphs. The main contributions of the proposed TESH-GCN is 1) incorporating textual information (extracted with language models) to node attributes 2) a sparse implementation of hyperbolic graph convolution. Empirical results on multiple real-world datasets suggest that TESH-GCN is competitive, efficient, robust and explainable.

**Audience:**

Yes

**Claims And Evidence:**

No

**Requested Changes:**

Revising the claims, adding discussions and experiments, and clarifying the unclear parts as stated in "Weaknesses".

**Strengths And Weaknesses:**

# Strength:
- The efficiency improvement of hyperbolic convolution seems novel and effective as it reduces a large amount of unnecessary computation (I am not an expert in hyperbolic graph neural networks, and thus this evaluation may not be strong).
- The experimental evaluations are extensive and cover many aspects of the proposed TESH-GCN, including performance, efficiency, robustness and interpretability. I specifically like the efficiency analysis (with respect to sparsity) and the explaining with meta-paths.

# Weaknesses:
## W1: There are some unjustified claims throughout the paper.
- In Introduction, the authors claim that "metapaths are only aggregated locally due to computational constraints". However, this is not accurate. For example, GTN (Yun et al. 2022) aims to learn meta-paths for heterogeneous graph neural networks. Also, they propose learning non-local meta-paths to complement GTN. Therefore, it is suggested to include GTN for discussion.

Graph Transformer Networks: Learning meta-path graphs to improve GNNs. In Neural Networks.

- In Introduction, the authors claim that "these aforementioned techniques fundamentally suffer from non-scalability when applied in practice". This is also not accurate. For example, PinSage (Ying et al. 2022) is a web-scale recommender system based on graph neural networks. HGT (Hu et al. 2020) is a heterogeneous GNN for web-scale graphs. Therefore, the authors should discuss these methods and revise the claim accordingly.

Heterogeneous Graph Transformer. WWW 2020

Graph Convolutional Neural Networks for Web-Scale Recommender Systems. KDD 2018

- In Introduction, the authors claim that "our unique integration mechanism ...". From my perspective, concatenating different features via residual connections is not unique. Many famous works (e.g. original Transformer) does so. It is OK to claim that "the aggregation brings robustness", but saying that it is unique is too strong.

## W2: Some technical contents are not clearly introduced.
- In Definition 2 (definition for hyperbolicity), the $dist$ function is not formally defined. Even if I take it as the graph edge distance, there are still unclear points. Specifically, in a heterogeneous graph, what kinds of edges are considered? Further, this definition is not easy to understand. Some examples may better illustrate the definition.

-  In Section 3.2, the authors claim to fuse the textual features into the adjacency matrix. However, this section is not easy to understand. I have several questions.
    - It is not clear why the modified $A$ will still contain adjacency information. For example, suppose node $i$ is connected to $j, k, l$, but $t_i[d] = 0$, then as in Eqn. 4, $A_k[d,i,:] = 0$, which indicates that the adjacency information between $i$ and $j, k, l$ is erased. Please correct me if there are misunderstandings.
    - It is not clear how this operation maintains the sparsity of adjacency matrices. In general, neural networks generate dense outputs, i.e. $t_i$ in general will not have values that are strictly $0$. Therefore, setting $A_k$ with $t_i$ intuitively compromises the sparsity, which seems to violate the following sections. Please clarify this point.

- Some notations are not used consistently. For example, in Section 3.1 $\mathcal{G}$ and $G$ are both used.

- In Eqn. 8-10, it seems that $h_L$ is a shared representation for all nodes in the same graph. This is counterintuitive as we are doing a node-level task. Any justifications?

## W3 Experiments are extensive but can still be improved.
- The authors are advised to introduce heterogeneous graph neural networks as baselines, such as HAN (Wang et al. 2019), MAGNN (Fu et al. 2020), GTN (Yun et al. 2022), and GraphMSE.

Heterogeneous graph attention network, WWW 2019

Magnn: Metapath aggregated graph neural network for heterogeneous graph embedding, WWW 2020

GraphMSE: Efficient Meta-path Selection in Semantically Aligned Feature Space for Graph Neural Networks, AAAI 2021

- The authors claim that the robustness is caused by the residual connection between text features and graph features. Therefore, in the ablation study in Fig. 9, the authors are advised to include variants w/o graph, w/o text into the comparison to better justify the point.

- For baselines such as GCN, GraphSAGE, HGCN, what are the input features? Please make it clear.

---

> ### Author Response · Authors · 2022-12-27
> **Revisions and Clarifications to Reviewer t96k**
>
> We are thankful for the reviewer’s extensive feedback and for noting the strength of our paper's novelty and experimentation. We hope the following responses address the questions and concerns raised by the reviewer. Please refer to the “Revisions” section to note the changes we will make following the review.
>
> Weaknesses:
>
> W1.1: We thank a reviewer for pointing us to the paper. It is definitely relevant. Revision 1.
>
> W1.2: The reviewer is right that there are scalable GNN architectures. However, they are not as capable of capturing hierarchies seen in several datasets as hyperbolic GNNs. In our paper, we propose a sparse tensor model to promote scalability in hyperbolic GNNs. This is non-trivial, because previous hyperbolic layers do not support sparse operations and this paper introduces the Sparse hyperbolic convolution layer for the same.
>
> W1.3: “the unique integration mechanism” here refers to adding semantic signals to the sparse adjacency tensor to retrieve metapaths from the graph structure. To the best of our knowledge, this is the first work that does this.
>
> W2.1: As the reviewer rightly assumed, the “dist” function refers to the edge distance between the nodes. In case of heterogeneous graphs, we do not consider the edge classes as any relation is sufficient to determine a boolean indication of edge. We apologize for the confusion.
>
> Revision 2.
>
> W2.2: Yes, the reviewer is right in the example that if $t_i[d]=0$, then the adjacency information will be removed. However, this is never the case practically, because $t_i[d]$ are taken from dense semantic vectors, which are never 0.
>
> W2.3: We apologize for the typos. We shall change all G to $\mathcal{G}$. Revision 3.
>
> W2.4:$h_L$ is computed for each edge and is not a shared representation for all nodes. The features are concatenated across the dimensions of semantic vectors. The entire computation of the sparse hyperbolic convolution is performed at an edge level.
>
> W3.1: In our experiments, we focused on the diversity of baselines that would perform the best on our datasets. The heterogeneous graph networks, proposed by the reviewer, would not be formidable baselines in our case, because they are more applicable when the relations have semantic meaning, e.g., $JoeBiden$\xrightarrow[\text{world}]{\text{is president}}UnitedStates$. Our datasets, on the other hand, have labeled classes, which are more under the scope of traditional GNN methods, or text+GNN methods. We will definitely discuss it as part of related work for completeness. Revision 1.
>
> W3.2: We refer the reviewer to Table 4, that includes results of our ablation study, including experiments on “w/o Text”, “w/o Hyperbolic”, “w/o Residual” and “CE loss”.
>
> W3.3: The input features are standard node features contained in the dataset, whenever present. In cases, when they are not present, we use fixed semantic embeddings. Revision 4.
>
> Requested Changes:
>
> Answered in Weaknesses.
>
> Revisions:
> 1. Added to related work: “Another line of work specifically tailored for heterogeneous graphs Fu et al. (2020); Yang et al. (2021); Hu et al. (2020),  Yun et al. (2019), and Wang et al. (2019) utilizes the rich relational information through metapath aggregation.”
> Added to references:
> “Yun, S., Jeong, M., Kim, R., Kang, J., & Kim, H. J. (2019). Graph transformer networks. Advances in neural information processing systems, 32.
> Wang, Xiao, Houye Ji, Chuan Shi, Bai Wang, Yanfang Ye, Peng Cui, and Philip S. Yu. "Heterogeneous graph attention network." In The world wide web conference, pp. 2022-2032. 2019.”
>
> 2. Added to Definition 2: “where dist(a,b) is the edge distance between nodes a and b in a homogenized version of graph $\mathcal{G}$”
>
> 3. Prev: “hierarchy of the graph G”, “hyperbolicity implies G“.
> New:“hierarchy of the graph $\mathcal{G}$”, “hyperbolicity implies $\mathcal{G}$“
>
> 4. Added to Section 4.2 Baselines: “In the case of graph-based methods, we utilize the node features provided in the dataset as default, else we utilize fixed-semantic vectors from the pretrained LM Song et al. (2020).”

---

> > ### Comment · Reviewer_t96k · 2022-12-28
> > **Followup question on Section 3.2 and Eqn. 4**
> >
> > I appreciate the authors for their reply. I will carefully check it.
> >
> > Before responding to the reply, I have a quick followup question on Section 3.2 and Equation 4.
> >
> > Suppose $i$ and $j, k$ are not connected, but $t_i[d]\neq 0$, which is what you say in the reply. Then, according to Eqn. 4, $A_k[d, i, :]\neq 0$, which implies that $A_k[d,i,j]\neq 0, A_k[d,i,k] \neq 0$.
> >
> > Then, if the above understanding is correct, it seems that the adjacency matrix will lose its sparsity. Please clarify it if I made any misunderstandings.

---

> > > ### Author Response · Authors · 2022-12-28
> > > **Author Reply to Followup question on Section 3.2 and Eqn. 4**
> > >
> > > We thank the reviewer for their prompt follow-up.
> > > The reviewer's question is great and they are right that Eqn. 4 will increase the density of $A_k$.
> > > However, the equation only adds one row and one column, which implies an increase of density in the order of $\frac{2|V|}{|V|\times|V|}=\frac{2}{|V|}$. From Table 2, we can note that the increase of $\frac{2}{|V|}$ will not significantly change the sparsity ratio of the datasets, and the adjacency tensors' sparsity will yet benefit from our model's design.

---

### Review · Reviewer_opHP · 2022-12-17

**Summary Of Contributions:**

The main contribution of this paper is TESH-GCN, a graph neural network which can learn metapath features and which employs a hyperbolic graph convolution layer. The proposed model is evaluated in the task of link prediction where it outperforms the baselines on most datasets. The authors also investigate the robustness of the model to noise and show that it is explainable.

**Audience:**

Yes

**Broader Impact Concerns:**

There are no concerns on the ethical implications of the work  that would require adding a Broader Impact Statement.

**Claims And Evidence:**

No

**Requested Changes:**

Most requested changes are related to the weaknesses listed above.

- In pages 2 and 3, the authors list some challenges. TESH-GCN is designed to solve those challenges. But the authors provide no explanations why the model can address those challenges. There are also no theoretical results that support their claims. In my opinion, more intuition on why the method is designed as it is and why it works on the employed datasets is necessary.

- Some claims made by the authors (described above) are not true in my view. The authors should rephrase the corresponding sentences.

- The proposed method is evaluated only in the task of link prediction. I would suggest the authors also evaluate the proposed model in other tasks such as in node classification.

- The comparison against some baselines is not fair since not all baselines can take the graph structure as input. For instance, BERT and XLNet can take only text as input. Furthermore, it is not clear to me whether in the case of GraphSAGE, GCN and HGCN, the nodes of the graph are annotated with features extracted from their textual content. If this is not the case, the comparison is not fair. I would suggest the authors make sure that the list of baselines consists of methods that take the same data as the proposed model as input.

- As discussed above, in some places, the paper lacks clarity. The authors should work on improving the presentation.

- I am not sure whether Table 6 is correct. Graph neural networks such as GCN use sparse representations of graphs and thus their complexity is linear in the number of edges.

**Strengths And Weaknesses:**

Strengths
--
- The authors conduct an extensive experimental validation of the TESH-GCN model. They perform an ablation study to investigate the importance of the different components of the model, while they also study its robustness to noise and also provide some examples to illustrate the model's explainability.

- The proposed TESH-GCN model achieves state-of-the-art performance in the task of link prediction. It outperforms all baselines on 4 out of 5 link prediction datasets.

Weaknesses
--
- The authors make several claims in the paper, but it is not clear to me if these claims actually hold. For instance, in the challenges listed in pages 2 and 3, the authors claim that there are no models that can aggregate graph-level metapath structures. However, such models exist (see [1]). They also claim that previous works do not leverage the complementary nature of graphs and text to improve robustness to noise. But there actually exist several models that combine graphs with text (nodes are usually annotated with their textual content). Finally, the authors claim that prior methods fix the semantic features of the nodes and do not allow the model to learn task specific embeddings. If I am not wrong TESH-GCN also uses a pre-trained language model to compute some fixed-size representations of the nodes' textual content, thus it also suffers from this problem.

- While the authors mention that the proposed model can be applied to other graph learning tasks (such as node classification), they only evaluate the performance of TESH-GCN in the task of link prediction. Link prediction has not been studied as much as other tasks in the past years, and thus the number of baselines is limited. On the other hand, in the task of node classification, there are several models that the authors could include in their list of baselines (just report the results from the original papers). I would thus suggest the authors consider other learning tasks along with link prediction.

- The paper lacks clarity and the whole section that presents the TESH-GCN model is not depicted well. Thus, the writing needs to be improved before the paper is ready for publication. There are several mathematically inappropriate descriptions. Some of them are easy to fix. For example, in page 8, it is not clear how exactly neighborhood aggregation is performed. It is mentioned that $x \in A_k$ but $A_k$ is a tensor. Furthermore, $A_k$ is three dimensional, while in Equation (5) $x$ has two dimensions. I suggest the authors work on improving the presentation.

[1] Fu, X., Zhang, J., Meng, Z., & King, I. "Magnn: Metapath aggregated graph neural network for heterogeneous graph embedding". In Proceedings of The Web Conference, pp. 2331-2341, 2020.

---

> ### Author Response · Authors · 2022-12-27
> **Revisions and Clarifications to Reviewer opHP**
>
> We thank the reviewer for their extensive feedback and for noting the strength of our paper's experimentation and results. We hope the following responses address the questions and concerns raised by the reviewer. Please refer to the “Revisions” section to note the changes we will make following the review.
> Weaknesses:
> - Metapath Structures: We request the reviewer to please note that our model utilizes “graph-level” metapaths. The approach [1] has been discussed in the paper and only uses “local” neighborhood metapaths because it does not use sparse tensors and relies on random walks. Our method encodes sparse adjacency tensors and hence is able to deal with long relations (Fig 1(a)). We have clarified this further with Revision 1.
> Integrating Text with Graph: There are two points to note in this case. The approach that the reviewer pointed out, same as TextGNN and TextGCN discussed in the paper, use fixed semantic embeddings to initialize GNNs. Our approach differs in two aspects; first it makes the pre-trained LM part of the end-to-end TESH-GCN model and the LMs are trained to optimize for the multi-step loss, and second TESH-GCN uses hyperbolic GNNs that are better at capturing hierarchy available in the datasets.
> - The reviewer is right that we do not experiment on node classification. However, we only added the footnote “Other tasks (node/graph classification) can be easily performed by changing the loss function” for help with adoption in the community. Link prediction is an extremely popular task in graph processing with several applications in domains of search and recommendation systems, medicine and e-commerce. We provide evidence of performance on several datasets against state-of-the-art baselines for the task of link prediction. However, if the reviewers find the footnote misleading, we will definitely remove it in the revision.
> - We apologize for the confusion. Revision 2 should help clarify it better.
>
> Requested changes:
> - Revision 1.
> - Revision 1.
> - Answered in Weaknesses 2.
> - The aim in our experiments was to compare the models against a diverse set of methods, and hence, we choose methods that are only-text, only-graph and text+graph. We have chosen TextGNN, TextGCN and Graphormer which are state-of-the-art text+graph methods. However, for completeness, we believe the comparison with only-text and only-graph methods is also necessary.
> - Revisions 1 and 2.
> - The reviewer is right that the basic formulation of GCN uses a sparse formulation. But practically, GPU machines do not support sparse formulations and hence GCNs need to be operated on as dense matrices (we used torch-geometric library) and hence their practical time complexity is $O(V^2)$, as given in Table 6.
>
> Revisions:
> 1. Previous: “...(TESH-GCN) to capture the graph’s metapath structures using semantic signals and further improve prediction in large heterogeneous graphs. In TESH-GCN, we extract semantic node information, which successively acts as a connection signal to extract relevant nodes’ local neighborhood and graph-level metapath features from the sparse adjacency tensor in a reformulated hyperbolic graph convolution layer…”.
> New: “..(TESH-GCN). In TESH-GCN, we use semantic node information to identify relevant nodes and extract their local neighborhood and graph-level metapath features. This is done by applying a reformulated hyperbolic graph convolution layer to the sparse adjacency tensor using the semantic node information as a connection signal.…”
> 2. Prev: “...For an input adjacency tensor with elements $x \in A_k[d]$,..”
> New: “...For the $d^{th}$ input adjacency matrix with elements $x \in A_k[d]$,..”

---

### Review · Reviewer_vw3S · 2022-12-19

**Summary Of Contributions:**

This work focuses on improving the hyperbolic GNNs using textual description. It proposes a method called TESH-GCN. It first extracts the local neighborhood information using textual information, and then it can better utilize the long-range metapath information. Finally, it aggregates metapath information using an attention module. The authors have shown comprehensive empirical results of TESH-GCN.

**Audience:**

Yes

**Claims And Evidence:**

Yes

**Requested Changes:**

## Requested Changes
- In the Abstract, the following two sentences have some overlap meanings.
  - `... to capture the graph’s metapath structures using semantic signals and further improve prediction in large heterogeneous graph.`
  - `we extract semantic node information, which successively acts as a connection signal to extract ...`
  - Also, the second sentence is too long to understand. The authors should polish it further.
- In Abstract, "... for the final downstream tasks."
- In Abstract, "... the ~~current~~ state-of-the-art approaches...".
- In Sec 1, can authors help explain this sentence: "However, they are only aggregated locally due to computational constraints"? Also, how does the metapath relates to the semantic information? Does semantic information belong to the metapath? An explicit discussion is helpful.
- In Figure 2, it seems that all the nodes have only one feature, which is the textual description. Can authors confirm this?
- Table 1 can be moved to the appendix since the key notations have already been given in the plain text.
- Typo in Sec 3.2, "... we detail ...".
- Also in Sec 3.2, I'm not convinced by the claim that "Each dimension of vector $t$ denotes a unique semantic feature ...", can authors give a more detailed explanation of this? For example, what is the exact meaning of each dimension? This is the most critical question in the algorithm design, because it reflects why separating each dimension is reasonable. Till now, without any proof, this claim is not rigorously valid.
- And following this, the authors claim that "thus, each feature needs to be added to a single adjacency matrix...". According to this context, "each dimension" is more explicit than "each feature".
- In Eq 4 and Fig 4, how is $A_k[d,i,j]$ defined? This is not well explained in the equation and figure.


**Strengths And Weaknesses:**

### Strengths
- This work is well-motivated.
- The authors consider comprehensive baselines.
- The authors explain the quantitative and qualitative results, which can better illustrate the robustness of TESH-GCN.

### Weaknesses
- Some notations or terminologies should be carefully introduced, such as metapath sturctures and semantic information. Currently, they are not well explained, thus the difference between this work and the existing ones is unclear. Specifically:
  - What's the difference between `nodes containing text` and `available semantic information`? Are these two the same?
- The writing can be further polished up, and some claims are not well supported. The detailed comments are in `Requested Changes` below.
- The technical novelty is limited.
  - The idea of incorporating the textual data into graph learning is interesting, but in terms of the method, TESH-GCN only conducts it as a way to build up adjacency tensors, which seems a little weak. Besides, the motivation for buidling D separate adjacent tensors is not well supported. Please check `Request changes`.
  - Nevertheless, the rich empirical analysis alleviates this issue to some extent. Just want to point this out to the authors & readers.
  - [OntoProtein](https://arxiv.org/abs/2201.11147) is a recent work in ICLR'22, and it's working on the protein-related KG, which is different from TESH-GCN. In addition to this, I'm wondering what's the difference from the technical viewpoint.
  - In Sec 1, authors claim that previous works are using fixed pre-trained textual representation. Meanwhile, OntoProtein adopts a pretrained LLM and applies an end-to-end pretraining on the heterogeneous graph.

---

> ### Author Response · Authors · 2022-12-27
> **Revisions and Clarifications to Reviewer vw3s**
>
> We are grateful to the reviewer for their thorough feedback and for highlighting the strength of our paper's motivation and experimental completeness. We hope the following responses address the questions and concerns raised by the reviewer. Please refer to the “Revisions” section to note the changes we will make following the review.
>
> Weakness:
> - Metapath Structures: We have briefly introduced metapath structures as a short description, Figure 1(a) and Challenge (1) in the introduction. We also referred the readers to the paper Fu et al. (2020) for additional details. However, we understand that a more elaborate description would help the readers further, and hence, we shall discuss it more in “Section 2: Related Work”. Semantic Information: They are the same. Sorry for the confusion. (Revision 1 and 5).
> - Revisions.
> - The idea of building D matrices is answered in Requested Changes (8). OntoProtoNet is a great work for protein networks, however, it optimizes the network on a translation loss (TransE). In our work, we need to consider different datasets and a link prediction+classification task to learn task-specific semantic embeddings and we design our model to use trainable semantic encoders optimized for the multi-step loss. We shall discuss the same in the paper’s related work.
>
> Requested Changes:
> - Revision 2.
> - Revision 3.
> - Revision 4.
> - Revision 6.
> - In Fig. 2, the nodes only have text information, which is encoded in a multidimensional semantic vector.
> - Revision 7.
> - Revision 8.
> - The claim is based on the i.i.d. Assumption in neural networks. While each semantic feature may not have an explainable meaning, it belongs to an independent dimension in latent space. Hence, we need D adjacency matrix to add to each dimension separately.
> - Revision 9.
> - Provided in the sentence below Eq.4: “where $A_k[d, i, :]$ represents the i th row in the d th matrix of $A_k$ and $A_k[d, :, j]$ represents the $j^{th}$ column in the d th matrix of $A_k$. $t_i[d]$ and $t_j[d]$ are the d th dimension of their respective semantic signals.”
>
> Revision:
>
>  Abstract:
> 1. Prev: “...nodes containing text…”.
> New: “...nodes containing semantic information…”
>
> 2. Prev: “...(TESH-GCN) to capture the graph’s metapath structures using semantic signals and further improve prediction in large heterogeneous graphs. In TESH-GCN, we extract semantic node information, which successively acts as a connection signal to extract relevant nodes’ local neighborhood and graph-level metapath features from the sparse adjacency tensor in a reformulated hyperbolic graph convolution layer…”.
> New: “..(TESH-GCN). In TESH-GCN, we use semantic node information to identify relevant nodes and extract their local neighborhood and graph-level metapath features. This is done by applying a reformulated hyperbolic graph convolution layer to the sparse adjacency tensor using the semantic node information as a connection signal.…”
>
> 3. Previous: “... for the final downstream task.”
> New: “... for the final downstream tasks.”
>
> 4. Previous: “... the current state-of-the-art approaches…”.
> New: “... the state-of-the-art approaches…”
>
> Introduction:
>
> 5. Prev: “...nodes containing text…”.
> New: “...nodes containing semantic information…”
>
> 6. Prev:”...to computational constraints. The adjacency tensor of a heterogeneous graph can be used to extract both metapath information as well as aggregate local neighborhood features…”.
> New: “...to computational constraints, i.e., only a local k-hop neighborhood of a heterogeneous graph’s node is considered while learning the metapaths. However, global metapaths can capture long-term relations between the nodes. To learn metapaths, we need to encode the path between two nodes and the semantic information contained in the path. Thus, The adjacency tensor of a heterogeneous graph with a semantic signal can be used to extract both metapath information as well as aggregate local neighborhood features…”
>
> Section 3: ​​ The Proposed model
>
> 7. Table 1 was left as a reference point to avoid any confusion. It has been moved to Appendix.
>
> 8. Prev: “...We detail…”.
> New: “...we detail…”
>
> 9. Prev:”...each feature…”.
> New:”...each dimension…”

---

### Decision · Action_Editors · 2023-02-13

**Recommendation:** Reject

**Comment:**

Two of the three reviewers recommended to reject the paper while one was leaning towards accepting the paper.

After going through the reviews and revisions I reach the decision to reject the paper mainly because all 3 reviews unraveled clarity issues which to me reflect that the paper is not yet completely ready to be published, a major revision is needed. This major revision should focus on the clarity of the exposition of the method. I encourage the authors to thoroughly go again through all the comments from the reviewers and think about how they can improve the manuscript (e.g. adding examples, clarifying some concepts, adding formal definitions, extending the discussions on key concepts used in the paper such as metapath, etc.) in order to avoid the points of confusion raised by the authors. (some points raised by the reviewers have been answered in comments but not addressed in the revision).

As an illustration, I will list a couple of points that I believe have not been properly addressed in the revision. This list is not at all exhaustive and I again encourage the authors to thoroughly go through the reviews and through the manuscript  before re-submission.

- The section "Incorporating Semantics into Adjacency Tensor" is not clear enough. Confusions on this aspect of the contribution have been raised by all 3 reviewers, however there were almost no changes made in the revision in this section. This section needs to be rewritten / clarified. E.g., the comments on sparsity from reviewer t96k has been answered in the reply from the authors but not clarified in the manuscript. Reviewer opHP suggested incorporating a simple example to illustrate the method here, this could be very beneficial. Note that this section needs to be significantly reworked (this is not just about adding one sentence or changing a term or an equation).

- Reviewer vw3s raised the need for a more careful definition of meta-paths. In my opinion the changes made in the revision to address this point are not sufficient. The notion of meta-paths is central to the claims and approach of the authors, yet no clear definitions of what a meta-path is is given. Here either a formal definition or a simple (but clear) example could help (or likely both).

- Reviewer vw3s raised a missing relevant discussion with the OntoProtein method. I did not see this point addressed in the revision.

- Similarly W1.2 W1.3 raised by reviewer t96k have not been addressed in the revision as far as I see. W2.1 from the same reviewer may be better addressed: is the notion of "homogenized version" of the graph clear / formally defined / unambiguous ? Here also an example may help.

- The point raised by reviewer opHP on the asymptotic complexity of GCN should be clarified in the manuscript (it has only been answered in the rebuttal as far as I see).


Other minor points I noticed:
- Eq 4 should be \forall d = 1 ... D instead of \forall d: 1 \to D
- Eq 2  :replace |E| by |V|^2 k (or mention near this equation that |E| =  |V|^2 k since E denotes the adjacency tensor here, in contrast with the standard usage where E denotes the set of vertices.
- In the experiment section, briefly clarify which methods are text only, graph only and text+graph and briefly motivate the choice of these baselines (as done in the reply to reviewer opHP).
- Table 1 is still in the main paper (in the reply to reviewer vw3s the authors mentioned it should be moved in appendix).



**Audience:**

Yes, this contribution is of interest to TMLR's audience, but the exposition needs to be clarified for the paper to be well understood by the community (which will also benefit its potential impact).

**Claims And Evidence:**

All three reviews reflect that some parts of the exposition of the proposed approach and architecture and claims are not clearly enough presented, this hinders the clarity of the claims made by this submission.